# Flash heating process for efficient meat preservation

Yimin Mao[1,2,6], Peihua Ma [3,6], Tangyuan Li[1,6], He Liu [1,6], Xinpeng Zhao [1], Shufeng Liu[1], Xiaoxue Jia[3], Shaik O. Rahaman[3], Xizheng Wang [1], Minhua Zhao[1], Gang Chen[1], Hua Xie [1], Alexandra H. Brozena [1], Bin Zhou[4], Yaguang Luo[4], Rodrigo Tarté [5], Cheng-I Wei [3], Qin Wang [3], Robert M. Briber [1] & Liangbing Hu [1] ✉

Maintaining food safety and quality is critical for public health and food security. Conventional food preservation methods, such as pasteurization and dehydration, often change the overall organoleptic quality of the food products. Herein, we demonstrate a method that affects only a thin surface layer of the food, using beef as a model. In this method, Joule heating is generated by applying high electric power to a carbon substrate in <1 s, which causes a transient increase of the substrate temperature to > ~2000 K. The beef surface in direct contact with the heating substrate is subjected to ultra-high temperature flash heating, leading to the formation of a microbe-inactivated, dehydrated layer of ~100 μm in thickness. Aerobic mesophilic bacteria, Enterobacteriaceae, yeast and mold on the treated samples are inactivated to a level below the detection limit and remained low during room temperature storage of 5 days. Meanwhile, the product quality, including visual appearance, texture, and nutrient level of the beef, remains mostly unchanged. In contrast, microorganisms grow rapidly on the untreated control samples, along with a rapid deterioration of the meat quality. This method might serve as a promising preservation technology for securing food safety and quality.

Meat is an important source of proteins, fatty acids, vitamins, as well as minerals such as iron and zinc, making it an important food source for human beings[1–3]. Global meat consumption has been growing steadily for several decades, exceeding 300 million tons since 2018[4]. However, meat is also prone to spoilage, due predominantly to the rich nutrient composition providing an excellent growth media for microorganisms, including food-born human pathogens and spoilage microorganisms. Thus, meat needs to be properly preserved and stored to ensure safe consumption.

Meat preservation is typically achieved by storing at low temperatures (4 °C within 4 h of slaughtering), or by freezing (<−20 °C) or super-chilling in a partially frozen state. In general, low-temperature methods have the advantage of maintaining better meat freshness compared to other techniques such as drying, smoking, and canning, or through chemical preservation (using salts, nitrites, acids, etc.), or biopreservation (using natural antimicrobial agents such as essential oils, nisin, and lysozyme, etc.)[5,6]. However, safely transporting meat requires energy to maintain the cold chain storage, which over long

[1]Department of Materials Science and Engineering, University of Maryland, College Park, MD 20742, USA. [2]NIST Center for Neutron Research, National Institute of Standards and Technology, Gaithersburg, MD 20899, USA. [3]Department of Nutrition and Food Science, University of Maryland, College Park, MD 20742, USA. [4]USDA-ARS, Food Quality and Environmental Microbial and Food Safety Laboratories, Beltsville, MD 20705, USA. [5]Department of Animal Science, Iowa State University, Ames, IA 50011, USA. [6]These authors contributed equally: Yimin Mao, Peihua Ma, Tangyuan Li, He Liu. ✉e-mail: binghu@umd.edu

distance can be subject to disruption, risking spoilage and contamination with harmful microorganisms.

In this work, we demonstrate an ultra-high-temperature flash heating (UFH) method that can efficiently preserve meats while retaining their nutrient level and texture. The concept stems from a similar ultra-fast and high-temperature heating method, also known as flash Joule heating, that has been previously applied in materials science to sinter ceramics[7], synthesize high entropy catalysts[8], recycle plastics[9], and convert waste (including food waste)[10], etc. Here, we propose to adapt this technology to food preservation. In this approach, Joule heating is achieved by applying a high electric power (~1800 W) over a short time interval (<1 s) to a carbon substrate, creating a transient increase of the substrate temperature to >~2000 K. By placing the meat in direct contact with the carbon substrate, this rapid heating causes ultra-fast dehydration and inactivation of microorganisms on the meat surface, forming a thin (~100 μm) protective layer that also serves as a barrier to inhibit inward bacterial migration toward the bulk of the meat. As a result of surface dehydration and microorganism-inactivation after the application of UFH, we achieved an extended shelf life of 5 days at room temperature for beef storage without compromising the interior appearance, texture, and nutrition of the meat. The UFH method is essentially a food surface treatment technology, which could be cost-effective for the industrial-scale preservation of large carcasses. The approach could be used alone or in combination with low-temperature technologies to decrease the risk of temperature variation during transport and storage for improved food safety.

## Results

### UFH creates a microbe-inactivated and dehydrated thin layer on the meat surface

A conceptual illustration of the UFH for the surface treatment of meat is shown in Fig. 1. The carbon felt is connected to a DC power supply (not shown), and is heated to >~2000 K in less than 1 s. For demonstration purposes, a beef steak (Fig. 1a) is placed on the carbon felt; the beef surface is subjected to ultra-fast dehydration and inactivation of microorganisms during the current pulse and associated high-temperature flash heating. The entire UFH process recorded by a high-speed camera (200 fps) can be seen in Supplementary Movie S1. Figure 1b compares UFH with other heating methods, in terms of temperature and operating time[11–17]. In general, the temperature in the UFH is an order of magnitude higher than that in conventional food heating methods, such as boiling, baking, and grilling; and it is more than 3–4 orders of magnitude faster. A butane torch may reach a temperature that is close to the UFH, but the heat transfer is mainly through convection, which is still 2 orders of magnitude slower. The effect of the UFH is schematically shown in Fig. 1c–e using a 1 cm³ beef cube. The heating procedure is applied to all six faces of the cube. Compared to the fresh beef (Fig. 1c), the UFH-treated beef possesses a thin,

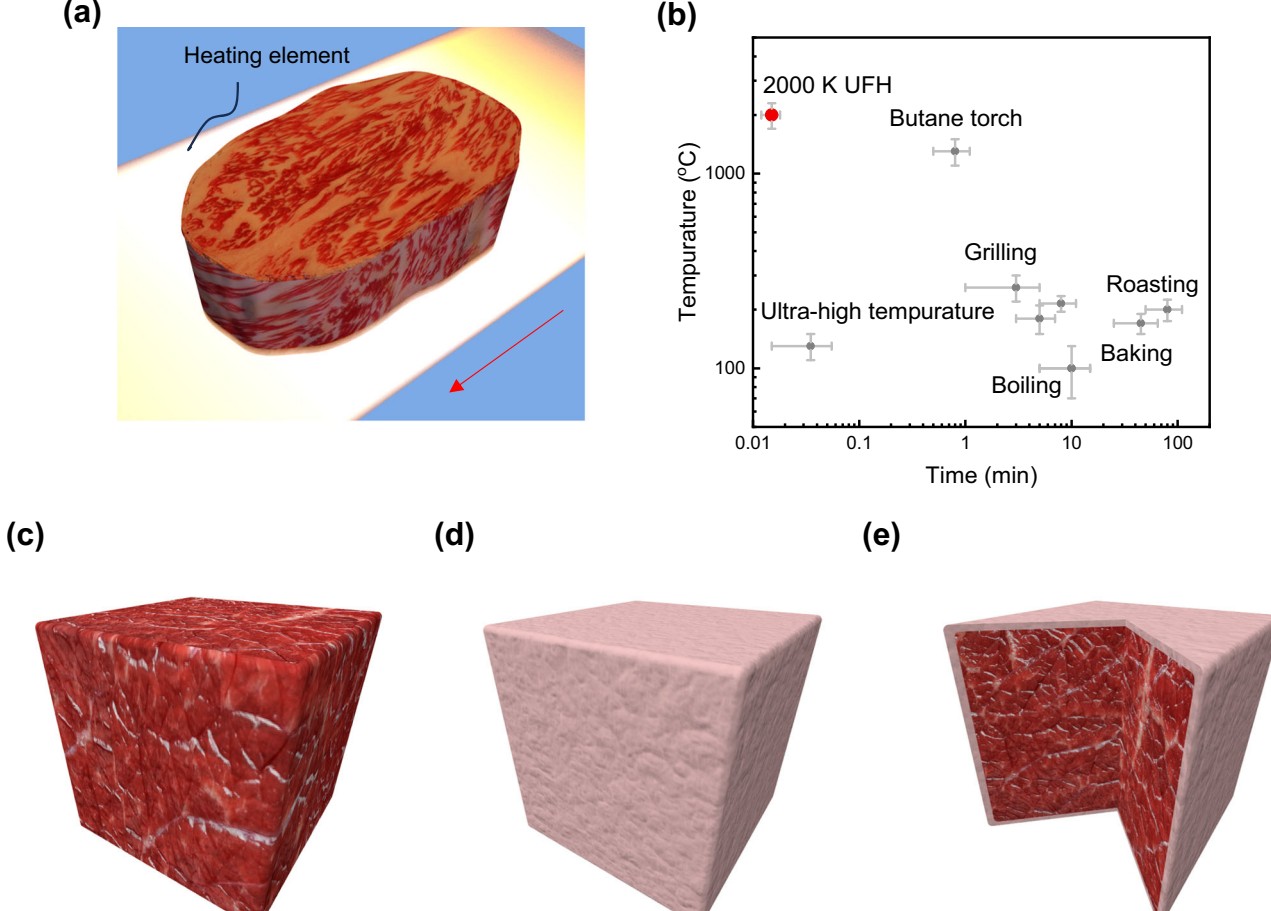

**Fig. 1 | Schematic of the UFH process for meat preservation. a** Demonstration of the heating process. Carbon felt is used as the heating element, due to its high electrical resistance. The arrow marks the electric current direction. **b** Comparison of operating temperature and time of conventional food heating methods with the UFH method (error bar symbols represent value ranges in the reported data). **c–e** Schematic appearance of a beef cube before (**c**) and after heating treatment (**d**, **e**). After the UFH is applied to the entire surface of the meat, producing a pale appearance (**d**), while the interior meat remains fresh, as revealed in the cross-sectional view of (**e**). The thickness of the outer layer with respect to the bulk sample is not to scale.

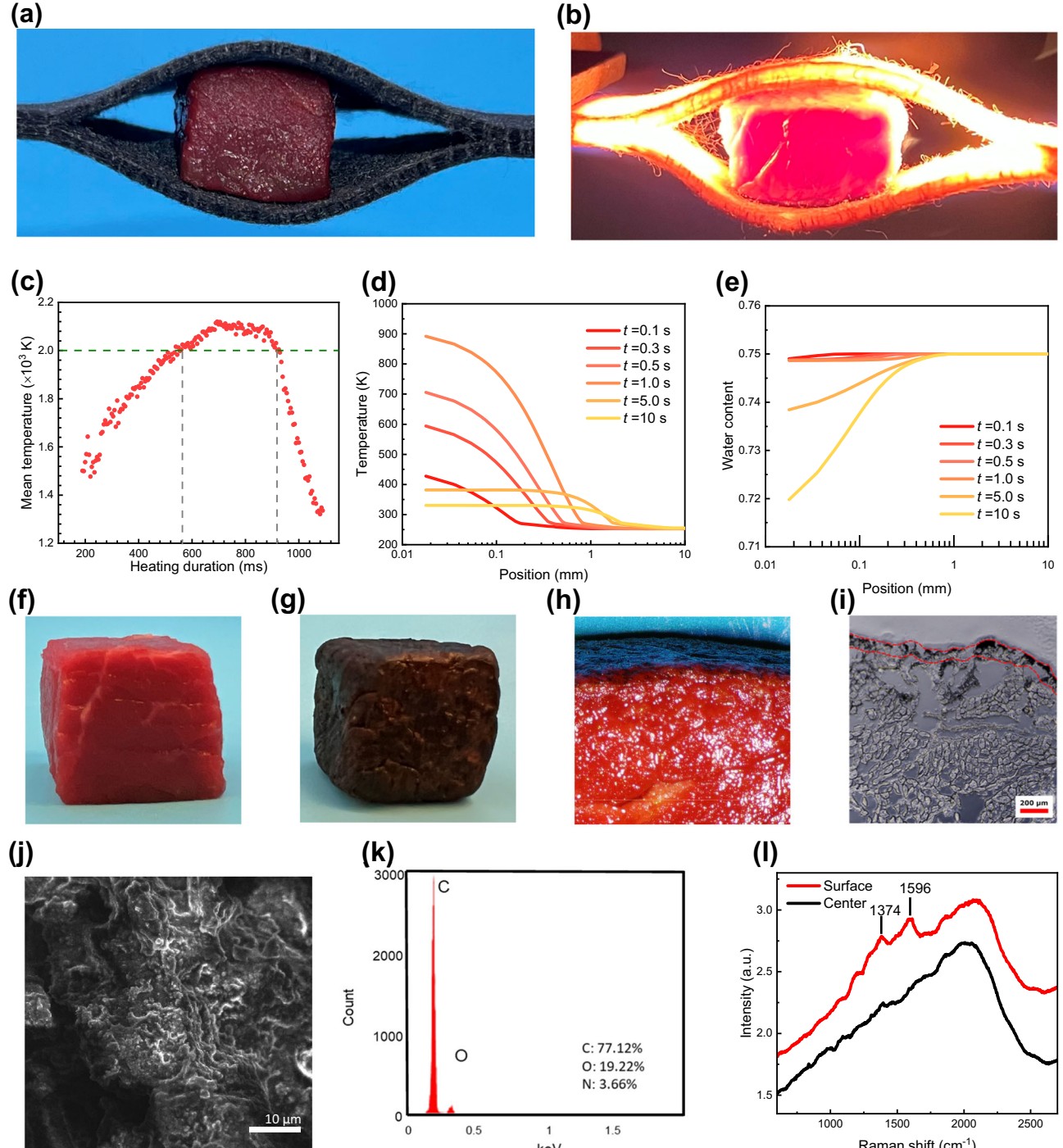

**Fig. 2 | Analysis of the UFH treatment of meat. a** Photograph of a beef cube of 1 cm³ sandwiched between two pieces of carbon felt that serve as the heating element. **b** The UFH treatment in operation. The light emitted by the carbon felts is due to the high temperature. **c** Temperature profile of the carbon felt during the UFH treatment. **d**, **e** Thermal analysis results of the evolution of the beef temperature (**d**) and water content (mass fraction) (**e**) over 10 s of the UFH treatment; the simulation sensors are along the central line normal to the carbon felt, from the heated beef surface across the entire sample. **f**, **g** Photographs showing a fresh (**f**) and UFH-treated (6 surfaces) beef cube (**g**). **h** Optical micrograph of the cross-section of a UFH-treated beef. **i** Histological micrograph of the surface of a UFH-treated beef (red line indicates the contour of the carbonized layer). **j**, **k** Scanning electron microscopy (SEM) image of the UFH-treated beef surface (**j**) and the corresponding energy dispersive spectroscopy (EDS) analysis (**k**). **l** Raman spectra of the surface and central part of the UFH-treated beef.

dehydrated outer layer that seals the surface (Fig. 1d, e); allowing the bulk beef to remain fresh (Fig. 1e).

As a proof-of-the-concept experiment, we placed a 1 cm³ beef cube (prepared from sirloin beef stored at ~253 K before use) between two pieces of carbon felt, which served as the Joule heating element (Fig. 2a). When a transient current passed through the carbon felts,

their temperature increased due to the Joule heating effect, causing the emission of electromagnetic radiation (Fig. 2b). The brightness of the carbon felt is associated with its temperature, which can be determined using a high-speed camera and color ratio pyrometry (Fig. S1)[18]. Fig. 2c shows a typical heating profile change for the carbon felt during the UFH process, where a temperature of ~1500 K at ~0.2 s

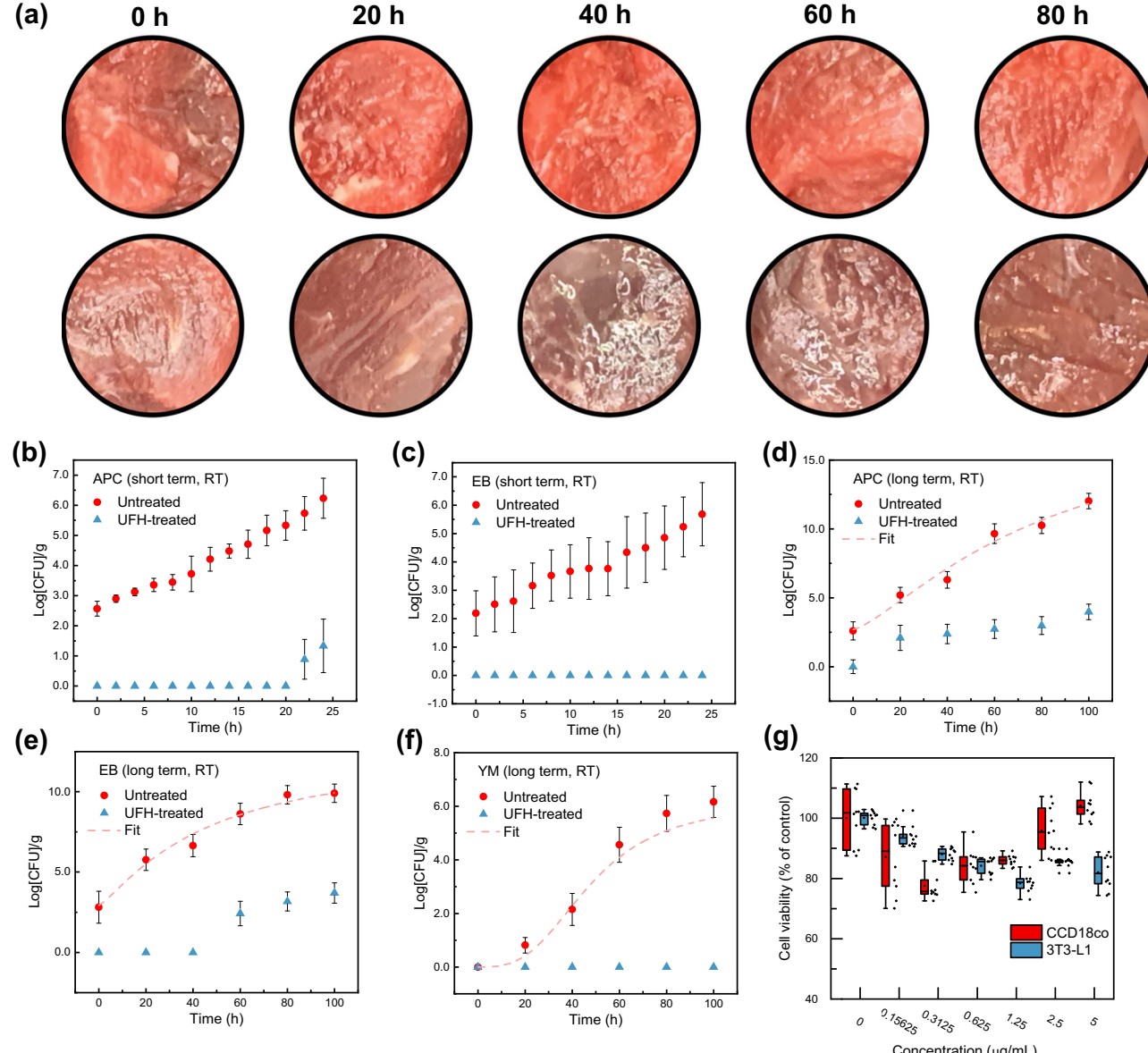

**Fig. 3 | Safety assessment of the untreated and UFH-treated beef. a** Photographs of the UFH-treated (top row) and untreated beef (bottom row) during storage. **b**, **c** Short-term (24 h) microbiological test for **b** APC and **c** EB. **d–f** Long-term (100 h) microbiological tests for **d** APC, **e** EB, and **f** YM. Solid lines are fits using the Baranyi–Roberts model. **g** Cell viability of 3T3-L1 (mouse embryonic fibroblast cell) and CCD18co (human colon tissue cell) at different concentrations of meat sampled from the surface of the UFH-treated beef. Current microbiological tests do not involve bacterial spores. Error bars indicate one standard deviation of uncertainty.

and a peak temperature of 2091 K at ~0.7 s was observed. The period during which the carbon felt temperature exceeds 2000 K lasts for ~0.2 s.

We conducted a two-dimensional numerical simulation to investigate the transient temperature distribution of the beef sample during the UFH process. To simplify the simulation, we assumed that only one face of the beef was in direct contact with the carbon felt heating element, with the other faces being exposed to an air environment at 293 K. The initial bulk temperature was set at 253 K, which is typical for long-term food storage[19]. The simulation results indicate that the beef surface temperature increased from ~450 K at 0.1 s to the peak point of ~900 K in 1 s during the UFH, then quickly dropped down below ~400 K in 5 s, after the heating was stopped (Fig. 2d). Our simulation also showed that at distances of >~2 mm from the heated surface, the beef temperature was almost unchanged.

The overall appearance of fresh and the UFH-treated beef, as well as the cross-section of the UFH-treated beef, are shown in Fig. 2f–h.

Image analysis (see Fig. S2 in the Supplementary Information for details) of the histological micrograph (Fig. 2i) reveals the thickness of the surface layer is ~100 μm. The dark appearance of the UFH-treated beef may be due to surface carbonization. Scanning electron microscopy (SEM) reveals the meat featured a rough surface after heating (Fig. 2j). Additionally, electron dispersion spectroscopy (EDS) analysis indicates that the dark regions consist mostly of carbon (Fig. 2k). Raman spectra reveal that the surface of the UFH-treated beef shows two enhanced peaks at 1374 and 1596 cm⁻¹ as compared with the central region of the meat sample (Fig. 2l). We assign these two broad peaks to the D- and G-band of graphite, indicating the existence of carbon species in a poorly ordered form[20,21].

**Microbiological tests reveal a significant inhibition in the growth of microorganisms during storage, due to the UFH treatment**

Meat appearance can be used as a quick check of the microorganism proliferation (Fig. 3a). The UFH-treated and untreated beef were stored

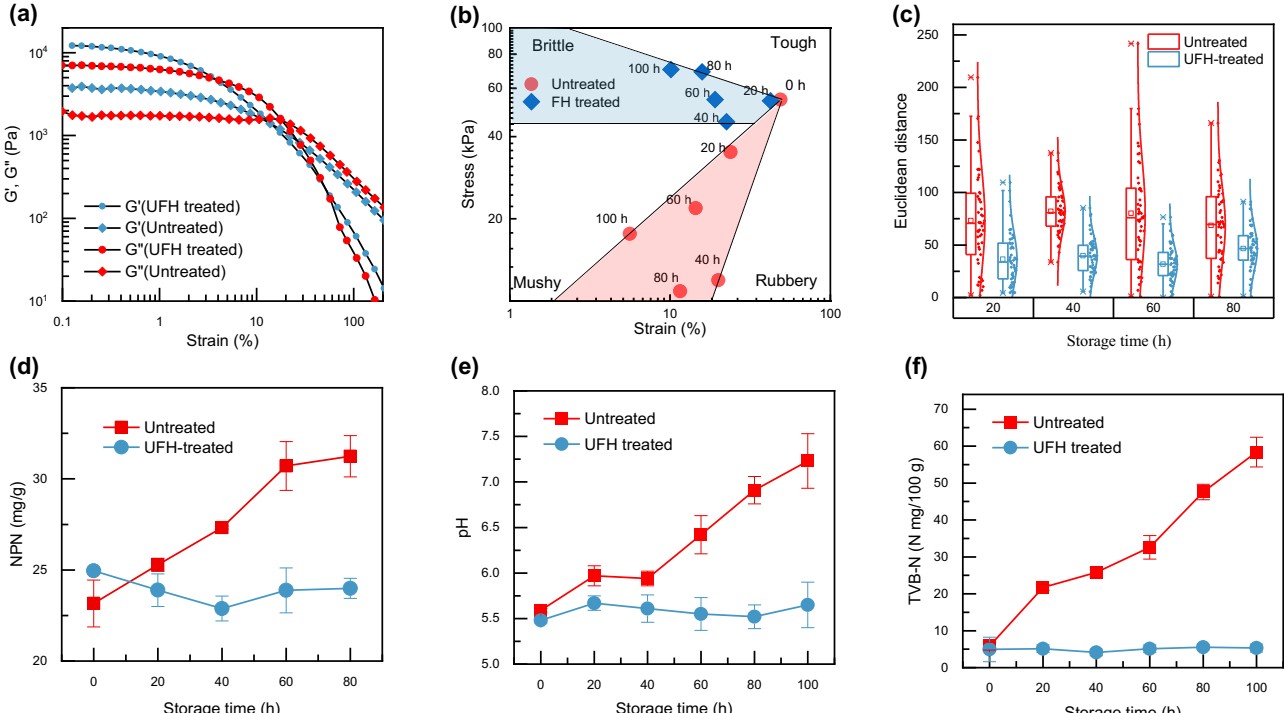

**Fig. 4 | Quality assessment of the untreated and UFH-treated beef. a** Rheological characterization based on the storage modulus $G'$ and loss modulus $G''$ as a function of oscillatory strain for the untreated and UFH-treated beef after 100 h of storage. **b** The texture change during storage. **c** Quantification of the beef color change during storage using the Euclidean distance. **d–f** Measurement of the **d** non-protein nitrogen concentration, **e** pH value, and **f** TVB-N concentration of the untreated and UFH-treated beef during storage. Error bars indicate one standard deviation of uncertainty.

at room temperature for 80 h, and their central regions were examined every 20 h. The visual appearance of the UFH-treated beef remained largely unchanged during storage under these conditions (top row in Fig. 3a). Meanwhile, the color of untreated beef became dark in the first 20 h and the surface appeared watery after 40 h due to microbial proliferation (bottom row in Fig. 3a).

To quantify the efficacy of the UFH method for inhibiting bacterial and fungal growth, we compared the content of microorganisms including total aerobic mesophiles (represented in aerobic plate count (APC)), *enterobacteriaceae* (EB), and yeasts and molds (YM), for the UFH-treated and untreated beef samples at room temperature (Fig. 3b–f). Microbiological tests were performed on a short-term (24 h) and long-term (100 h) basis; and the Baranyi–Roberts model was used to fit the microbial growth profile (solid lines in Fig. 3d–f)[22]. For the untreated beef, APC and EB counts increase rapidly in the first 24 h while the APC counts remain below the detection limit in the first 22 h and EB 24 h for the UFH-treated beef (the lower detection limit of APC and EB are 0.70 and 0.48 log CFU/g, see Methods section for details), as shown in Fig. 3b, c. Long-term tests reveal the complete microbial growth profiles for untreated beef (Fig. 3d–f for APC, EB, and YM, respectively). Note that data of short-term test for YM is not shown, as the YM count remains undetectable for both untreated and UFH-treated beef; but long-term test clearly shows a rapid increase in YM populations in the untreated beef after 20 h and reaching ~$10^6$/g (Fig. 3f) after 100 h. In contrast, no YM were detected in the UFH-treated beef during the entire 100 h period. At the late stages of storage, the untreated beef samples had completely spoiled, whereas the bacterial counts in the UFH-treated beef samples barely reached the critical limits[23]. For example, according to the USDA Agricultural Marketing Service (AMS), the APC critical limit for ground beef is 4.0 log CFU/g[24]. The APC counts of untreated beef stored at room temperature exceeds the critical limit after ~12 h (Fig. 3b), while that

of the UFH-treated beef remain below the limit even after storing for 100 h. We also conducted the cytotoxicity assessment of the surface of the UFH-treated beef to address potential safety concerns. Compared to untreated beef, the cell viabilities of 3T3-L1 (mouse embryonic fibroblast cell) and CCD18co (human colon tissue cell) did not change when cultured with the addition of beef sample collected from the UFH-treated surface (Fig. 3g).

## UFH treatment maintains beef quality during storage

Besides assuring safety, it is critical for a preservation method to maintain food quality as characterized by appearance, texture, and nutrient level. Texture is an important measure for meat quality, being directly associated with the mouthfeel during consumption[25]. We conducted rheological experiments including a strain sweep of the storage modulus $G'$ (elastic behavior) and the loss modulus $G''$ (viscous behavior) to assess how the UFH treatment might affect the texture of the beef samples. Rheological data of untreated and UFH-treated beef after 100 h of storage at room temperature is shown in Fig. 4a (rheological data was collected every 20 h during the storage, see Fig. S8). The untreated samples show $G'$ and $G''$ values that are ~4 times lower than the UFH-treated samples. Food texture can be generally categorized into four classes: tough, rubbery, mushy, and brittle as quantified by stress and strain values at the end of the linear viscoelastic region in a stress-strain curve[26,27]. Fresh as well as UFH-treated beef behave as a "tough" material (Fig. 4b). During storage, the texture of the untreated samples experienced a change from "tough" to "mushy" due to myofibrillar protein decomposition caused by microbial spoilage. The texture change of the UFH-treated beef is much more localized in the stress-strain plot (Fig. 4b). A slight shift towards the "brittle" region is due to moisture loss during storage at room temperature. These results show the UFH treatment has a negligible effect on the texture of the beef samples, which we attribute to the fact the process affects only the surface of the meat.

We also studied the difference between untreated and UFH-treated beef in terms of their freshness and nutritional value. The color of meat is a direct indicator of its freshness. We use the Euclidean distances of the RGB components of the color at randomly picked pixels on a meat photograph to characterize the appearance of the beef samples[28]. Using fresh carcass as a reference, the central region of the UFH-treated beef maintains at the same level of the Euclidean distance over 80 h, while the untreated beef group shows a significant increase in the Euclidean distance, which is also evidenced by the dark hue of the beef sample. This is due to various oxidation processes when meat is exposed to air and light[29–31], as well as the meat spoilage during storage (Fig. 4c)[6,30].

Proteolytic activities of the microorganisms break down the proteins in meat, creating peptides and amino acids that are further decomposed into low-grade amines and ammonia[32]. These deterioration processes raise the risk of food poisoning and may significantly lower the nutrient level. Three chemical markers, i.e., the non-protein nitrogen (NPN) content (Fig. 4d)[33,34], pH value (Fig. 4e)[35,36], and total volatile basic nitrogen (TVB-N) content (Fig. 4f)[37–39], were used to assess the freshness and nutrient level. All three markers remained essentially unchanged during storage for the UFH-treated beef. However, for the untreated beef, the NPN content increased from ~24 mg/g to ~31 mg/g by the late phase of storage at 80 h (Fig. 4d). Additionally, the pH value increased from 5.6 ($\pm$0.02) to 7.2 ($\pm$0.3) (Fig. 4f); and the TVB-N content increased by ~10-fold (Fig. 4f). These results strongly suggest that the UFH-treated beef can remain fresh for a time frame of approximately 100 h.

## Discussion

We attribute the marked inhibition of microbial growth after UFH treatment of the beef to the combined effects of surface dehydration and microorganism inactivation. The meat surface temperature reaches ~900 K due to the carbon felt temperatures of ~2000 K during the UFH treatment (Fig. 2d). More importantly, UFH causes water evaporation, leading to a dehydrated layer of ~1 mm in thickness (Fig. 2e), which is consistent with the surface water activity ($a_w$) measurements: $a_w$ at surface maintains at ~0.8 during storage for 80 h, while the untreated beef shows a much higher water activity of ~0.99 (Fig. S5). $a_w$ is one of the major parameters that limits food microbial growth. Most bacteria and many yeasts proliferate when $a_w > 0.91$ while the growth of most molds requires $a_w > 0.80$[40,41]. Maintaining a value of $a_w$ of ~0.8 at the beef surface plays a critical role in inhibiting microbial growth. These findings suggest that the UFH results in similar microbial growth inhibition mechanisms as conventional drying and dry-curing[42], but the latter methods significantly modify the overall meat texture and flavor.

It needs to be pointed out that thermal radiation is the dominating heat transfer mechanism in the UFH method, as evidenced from the high light emission of the heating element (Fig. 2b). As a result, the UFH treatment does not require perfect contact between the meat surface and the heating element, which allows it to sterilize the material even when the meat surface is rough. Additionally, the UFH method can be scaled up since the heating element is flexible and can effectively cover expansive areas with irregular surfaces, such as large carcasses, with imperfect contact being overcome by the radiative heat transfer.

Our objective of food processing is to preserve the freshness of food products, maintaining their native properties to as high a degree as possible without introducing chemical additives and/or prolonged processing time. The UFH method is a food surface treatment, producing a protective thin layer (sterilized and dehydrated) that helps delay various oxidation processes that may alter meat color and texture. Conventional heating methods, due to prolonged heating time (Fig. 1b), may significantly denature proteins. For example, beef myosin, actin, and collagen denature between 54–58 °C, 71–83 °C, and 65–67 °C, respectively[43,44]. As seen in Fig. 2d, temperature variation

only takes place on the meat exterior (~2 mm from the surface); and the interior temperature remains unchanged throughout the process. When heating is ceased, the inward heat transfer from the meat surface relies solely on thermal conductance, and the heat is quickly dissipated to the environment via convection (Fig. S10). In this regard, the UFH meat surface treatment is fundamentally different from conventional cooking, in which the entire piece of meat is cooked. Note that at this stage the UFH serves as an auxiliary meat preservation method since frozen meat was used as a model; however, no significant obstacles are foreseen to its application to fresh meats.

Understandably, concerns about potential health risks due to the extremely high-temperature treatment of the meat may arise. Therefore, we conducted a cytotoxicity assessment that indicated the UFH-treated surface does not show cell toxicity, since surface carbonization might be the dominant chemical process, as revealed by Raman spectroscopy (Fig. 2l). Additionally, the generation of acrylamide, a carcinogenic compound to humans, is one of the major health concerns of cooking methods such as grilling and frying. High-performance liquid chromatography-mass spectrometry (HPLC-MS) analysis indicates that the acrylamide concentration remains at a level of <7 parts per billion (ppb) in the UFH-treated beef surface (Fig. S3), which is far less than the 500-ppb benchmark limit set by the European Union regulation[45]. Furthermore, HPLC full scan analysis indicates that heterocyclic aromatic amines (HAA) and benzo(a)pyrene are absent from the samples (Tables S1 and S2).

The 'charred' appearance of the UFH-treated meat may impact the acceptance by customers. Therefore, the UFH method may be most useful for large-scale meat treatment (e.g., handling carcasses at slaughterhouses) since the cost-effectiveness is inversely proportional to the specific surface of the bulk meat. After preservation and transport, the 'charred' surface layer (~100 μm in thickness) can be removed from the bulk meat at the retailer's end to ensure that the product is appealing to consumers. Future research may be conducted to avoid the carbonization effect by lowering the heating temperature and still maintaining the formation of the dehydrated surface layer.

Economic gain beyond reduced energy usage due to lowered refrigeration needs may be achieved, owing to the room temperature microbial inhibition effect of the UFH. An increase in the storage temperature of food can be caused by unpredictable factors such as power outages or equipment malfunctions[46]. An extension of shelf life at room temperature may significantly mitigate the risk of food spoilage due to temporary unplanned temperature increases. A full life cycle analysis, however, is not available, because it depends heavily on multiple factors, such as the application targets (though in principle this method is most energy efficient for large meat pieces such as carcasses, smaller products such as primal or subprimal cuts of meat are other options) and meat distribution scheme, etc. The heating device needs to be optimized to satisfy various application needs.

## Methods

### UFH experiment

Carbon felt of 4 × 2.5 cm was used as the heating element (AvCarb G280A, FUELCELL Store). The two ends of a carbon felt were attached to graphite plates using copper clamps connected to a DC power source (MP10050D, StarPower). The output current and voltage of the DC power source were adjustable between 0–50 A and 0–100 V, respectively. The heating cycle time can be controlled. In a typical UFH experiment output power of ~1800 W was applied to a carbon felt in <1 s, corresponding to a temperature profile maintaining a peak temperature of ~2100 K for ~0.2 s (Fig. 2c). Two carbon felts can be used together to simultaneously heat the top and bottom surfaces of an object.

The temperature evolution of the carbon felt was measured using a high-speed camera (Vision Research Phantom Miro M110) based on color ratio pyrometry, with videos recorded at 200 frames/s[18].

The beef was purchased from a local retailer. To facilitate cutting and shaping, the beef was briefly stored in a freezer and then taken out and cut into cubes of 1 cm³. The beef cubes were then separated into two groups (used for the UFH treatment and control), each contained in a Ziplock bag, and were stored at −253 K until the UFH experiments.

## Thermal and water content analysis during UFH

We use COMSOL MultiPhysics software, based on the finite element method, to study heat transfer and the change in the water content of frozen beef during the UHF process. Our model, illustrated in Fig. S10, is reduced to a two-dimensional problem, where only the surface of the frozen beef is heated, while the remaining parts are exposed to ambient conditions. The heating element is positioned at the bottom of the frozen beef, generating a high temperature of 2000 K for a duration of 1 s. This heat transfer increases the beef temperature, melts the ice, and causes water within the beef to evaporate. The simulation is performed in a time frame of 10 s, including both the heating and the cooling phases. Details such as heat transfer equations, basic assumptions, boundary conditions, as well as material property parameters used in simulation are given in the Supplementary Information.

## Microbiological tests

The UFH-treated and control samples, 5 g each, were collected, immersed in 45 mL of sterile buffered peptone water (BPW), and stomached for 2 min using a stomacher (Seward type 80, UK). Additional four dilutions of $10^{-1}$, $10^{-2}$, $10^{-3}$, $10^{-4}$, and $10^{-5}$ were successively prepared. For the long-term test, five additional dilutions of $10^{-3}$, $10^{-6}$, $10^{-9}$, $10^{-12}$, and $10^{-15}$ were prepared.

*Aerobic Plate Count (APC)*. All dilutions were evenly dispensed onto the APC plate (Petrifilm™, 3M). The prepared plates were incubated at $35 \pm 1$ °C for 48 h, and the colony number was then counted. The countable range was between 25 and 250 CFU.

*Enterobacteriaceae (EB)*. All dilutions were plated following the same procedure for APC, using (Petrifilm™, 3M). The EB count plates were incubated at $35 \pm 1$ °C for 24 h, and the colony number was then counted. Red colonies associated with gas bubbles or colonies surrounded by yellow zones with or without gas were counted. The countable range was between 15 and 100 CFU.

*Yeast and Mold (YM)*. All dilutions were plated following the same procedure for APC, using rapid yeast and mold count plate (Petrifilm™, 3M). The EB count plates were incubated at $25 \pm 1$ °C for 48 h, and the colony number was then counted. All red colonies were counted regardless of their size or color intensity. The countable range was between 10 and 100 CFU.

## Chemical characterization

*Total protein content*. A nitrogen determinator (Leco TruMac N) was used to quantify the total protein content in beef samples. $1.0 \pm 0.2$ g of beef sample was loaded into a ceramic boat and was dried at $101 \pm 1$ °C for 45 min in a convection oven before the nitrogen analysis measurements. Blank boat weight was measured with a standard deviation of <0.002%. Ethylenediaminetetraacetic acid (EDTA) was used as a calibration standard. The furnace temperature was 1100 °C, with a lance flow of ≈1.8 L/min and purge flow of ≈4.2 L/min. A nitrogen-to-protein conversion factor of 6.25 was multiplied to report the protein content based on the measured nitrogen content.

*Total volatile basic nitrogen (TVB-N)*. The Conway method was used to determine TVB-N. 10 g of minced beef, with 100 mL of DI water added, was homogenized for 30 min (Ultra-Turrax T25, IKA, Staufen, Germany).

Into the inner chamber of a Conway dish with a diameter, 3 mL of boric acid absorption solution (20 g/L) and 50 mL of pH indicator (mixture of methyl red and bromocresol green at 1:5 volume ratio) were added; and into the outer chamber 3 mL of potassium carbonate saturated solution and 1 mL of filtered liquid beef homogenate were added, successively.

The Conway dish was then sealed and incubated for 2 h at 37 °C. HCl (0.1 mol/L) was used to titrate against the boric acid-absorbed nitrogenous compounds. The TVB-N content $c_{TVBN}$ (mg/100 g) was quantified by

$$c_{TVBN} = \frac{(v_1 - v_2) \times c \times 14}{m \times 0.05} \times 100, \tag{1}$$

where $v_1$ is the HCl titration volume for the tested sample (mL), and $v_2$ of the blank sample (mL); $c$ is the actual concentration of HCl (mol/L); $m$ is the weight of the beef sample (g).

## Cytotoxicity assessment

3T3-L1 and CCD18co cell lines were obtained from the American Type Culture Collection (ATCC, CL-173). Cells were seeded and cultured in the Dulbecco's modified Eagle's medium, containing sodium bicarbonate (3.7 g/L), penicillin-streptomycin (1%), and bovine calf serum (10%). Adipocytes were developed after treating the post-confluent cells with fetal bovine serum (10%), 3-isobutyl-1-methylxanthine (0.5 mM), dexamethasone (1.0 mM), and insulin (1.67 mM) after two days.

## Physical characterization

*Scanning electron microscopy (SEM)*. Beef samples were freeze-dried before the SEM (Tescan XEIA FEG SEM, Brno, Czechia) experiments. The sample surface was coated with a platinum layer of 1 nm thick; and the SEM measurements were performed under 10 kV accelerating voltage. A 4pi Analysis System was used to acquire the images (Durham, NC).

*Raman spectroscopy*. Freeze-dried beef samples were used for the experiments, using a Horiba Jobin Yvon (Edison, NJ) confocal Raman spectrometer. The excitation laser wavelength was 532 nm and the objective lens magnification was 10×. Spectra were collected using a 600 g/mm grating. Neutral density filters were added to prevent sample damage if needed.

*Texture analysis*. A strain sweep was performed on the beef samples with a thickness of 2 mm at room temperature, using a rheometer in parallel plate geometry (TA Instruments). Plateau stress and strain at the end of the linear viscoelastic region were used to quantify the beef texture.

*Histological microscopy*. Beef samples were soaked in phosphate buffer (pH = 7.4) containing 4% (volume fraction) of formaldehyde first, and were then plunged into liquid nitrogen-cooled isopentane. A cryomicrotome (Leica Microsystems 3050S, Nussloch, Germany) was used for sectioning. Histological studies were performed using an optical microscope (Olympus VS-BX), and micrographs were registered by a digital camera (VC50).

*Color analysis*. Euclidian distances $D_E$ were computed based on 50 randomly picked pixels on a photograph of beef, using the following equation:

$$D_E = \sqrt{(R_2 - R_1)^2 + (G_2 - G_1)^2 + (B_2 - B_1)^2}, \tag{2}$$

where $R$, $G$, and $B$ represent the three components, red, green, and blue, of the color at a given pixel. The subscripts (2 and 1) refer to two different pixels[28].

## Data availability

Source data are provided with this paper.

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

## Acknowledgements
LH acknowledges the support from the University of Maryland A. James Clark School of Engineering. The identification of any commercial product or trade name does not imply endorsement or recommendation by the National Institute of Standards and Technology.

## Author contributions
L.H. directed the project. Y.M. and L.H. conceptualized the invention. Y.M., P.M., T.L. and L.H. designed the experiments. P.M., T.L., Y.M., S.L. and G.C. carried out flashing heating experiments. P.M., X.J. and B.Z. performed microbiological tests. H.L., X.Z. and Y.M. carried out thermal analysis. S.O.R. performed food toxicity tests. P.M. carried out

microscopy and chromatography experiments, and meat texture analysis. Y.M. and M.Z. carried out spectroscopic measurements. P.M. and Y.M. carried out histology study. H.X. and X.W. performed high-speed video shooting. R.T. advised the application of the UFH method to meat products. Y.M., P.M., H.L., A.H.B., Y.L., R.T., C.-I.W., Q.W., R.M.B. and L.H. conducted collaborative writing.

## Competing interests

The authors declare no competing interests.
