## [Peer Review File · Nature Communications]

Flash heating process for efficient meat preservationREVIEWER COMMENTS

Reviewer #1 (Remarks to the Author):

Manuscript ID: NCOMMS-23-38956A-Z

Title: 2000 K Flash Heating for Highly Efficient Meat Preservation

Overview:

The article discusses the treatment of beef with ultrafast flash heating (UFH) to maintain the texture, quality, nutrition level, and visual appearance to increase the shelf life. The authors demonstrate that UFH can withstand bacterial spoiling for 5 days when stored at room temperature. The beef cube that was exposed to UFH undergoes some graphitization ($\sim 100 \mu\text{m}$ in thickness), but the rest of the meat is intact. The authors were able to control the heating time from 100 ms to 1400 ms to optimize the heating period that can yield the longest shelf life for the UFH-treated meat. The microbe analysis along with cytotoxicity states that the bacterial infection did not occur for 100 h. The interpretation of G' and G'' modulus indeed supports their claim that the meat quality was not changed after UFH treatment. Although the concept is very simple, it is effective and worth publishing in this journal. Here is a list of questions that must be considered. The reviewer is especially concerned on how they conducted the control experiment and how applicable this technique is to the food market industry and to a general household. Please see below.

1. Many chemical reactions (e.g., Maillard reaction) can occur during the heating process. Other than graphitization, what other chemical reactions occurred on the surface of the beef cube? Could the authors conduct more detailed chemical analysis to confirm the nature of surface after UFH? This is important to consider because we do not know whether the final product has some unwanted byproducts that might be detrimental to the consumer. For example, would it be possible to provide HPLC data of the surface of the beef cube after acid digestion?
2. The comparison between the control and the UFH-treated meat is not a fair comparison. The authors directly used the meat that was stored at $\sim 253 \text{ K}$, a normal freezer temperature. The chance that this meat was already in contact with bacteria is highly probable as there are no detailed experimental procedures that deal with the level of hygiene in preparing, cutting, processing, and shaping this meat into a cube. The control meat should also be pasteurized or treated once again to eliminate the germs before conducting the experiment.
3. How is this technique more advantageous than a common cooking technique like sous vide or any non-destructive/non-chemical preservation methods (vacuum sealing)? For example, in sous vide, the meat is immersed in a water bath (about 50-70 degrees Celsius) to eliminate the bacteria and improve the texture. Of course, UFH technique might be faster, but it is impossible to incorporate this technique into a normal house since it requires a N_2 -filled glove box and extremely high voltage. Or is this better for the factory/butcher shop, and not for homes?
4. The beef cube image shown in Figure 2h looks charred. Although the authors claimed that cutting the UFH-treated part, this might not be an attractive solution for the regular customer. For me, when I am buying a steak, I would be more inclined to buy the beef shown in Figure 2g.
5. The life cycle analysis of UFH treatment with a common pasteurization technique is required. Please include both refrigeration technique and vacuum sealing with subsequent heating to kill the germs.

Reviewer #2 (Remarks to the Author):

In general the manuscript is clearly written and appears to follow guidance on design and use of citations/references, as well as flow from the introduction to results, ending with Materials and Methods.

The key findings are the microbiological and the color data of the treated beef, though the data are limited by lack of assessment directly of inoculated pathogenic microbes (e.g., Salmonella, E coli), thermo-tolerant/extreme-thermotolerant or otherwise, or the assessment of consumer

liking/palatability of treated beef.

The data should be significant, given its innovation and novelty. The authors do a good job at comparing to existing technologies, but the manuscript would benefit from similar comparisons for other foods (e.g., poultry, minimally processed produce, etc.).

The methods appear to be sufficiently detailed to allow replication with sufficient research expertise.

The reviewer would suggest the results be revised to shorten the first page's text to more quickly initiate the discussion of gathered data and its interpretations, versus the early text that does not move the reader into evaluation of findings.

Reviewer #3 (Remarks to the Author):

Comments to the authors:

The manuscript shows an interesting and novel approach to preserve meat (and possibly other food products) by means of flash heating. There are some major concerns related to the methodology and to the research itself, which should be properly addressed before the manuscript reaches publication.

The main issue is related to the methodology itself. Beef in particular and food in general does not show regular surfaces. Plenty of protuberances, holes, crevices or even porous surfaces can be found in any piece of food, which will not easily adapt to the carbon felt. If, as authors claim, the flash heating method only reaches 0.1 mm depth, then microorganisms present in this sort of hiding places could find protection and escape the treatment. If the heated surface reaches deeper, then carbonization may be too extensive and may harm too much food quality. In one way or another, the proposed technology may be of little use for the food industry.

There are some other major concerns and minor remarks that authors should address.

Major concerns:

- 1.- Figure 2d shows the temperature profile of the carbon felt during the treatment. It starts measuring the temperature at about 1500 K and stops measuring, after the treatment, at about 1300 K. Both of them are extremely high temperatures. What does it happen before and mainly after the measured interval? Probably the increase in temperature is very fast, but usually cooling is not so fast and, from the plot, a "tail-kind" behaviour could be predicted, i.e., temperature could remain very high for a long time. And this is for the carbon felt. What about the piece of meat?
- 2.- Figures 3d and e: authors show growth curves reaching 10 log CFU/g or even more but have performed only 5 dilutions (line 323) and have countable ranges up to 250 CFU/plate (line 326). This means that the maximum log count they can reach is about 3×10^7 CFU/g.
- 3.- Carbonization: authors claim that carbonization might impact the acceptance of consumers. Actually, the piece of meat shown in figure 2h looks like quite unpleasant to me. Authors argue that, to avoid such carbonization effect, the heating temperature could be lowered. Then, why did not they try with a lower temperature in this very research?
- 4.- Cytotoxicity: authors test cytotoxic effects of their treatment but it would be probably more advisable to measure the formation of Maillard reaction compounds, such as acrylamide.

Minor remarks:

- 1.- The objective of the research is not clearly stated. Although it can be easily drawn from the context, it should be properly written.
- 2.- Figure 1a: it does not show the apparatus and processing, but a piece of meat.
- 3.- Figure 1e: if the cube is 1 cm³ volume, then the picture roughly shows a surface layer of almost 0.5 mm width, not 0.1 as claimed by the authors.
- 4.- Lines 91-92: how is the treatment applied to the six faces of the cube? Is it a discontinuous process that has to be applied in three different times?

- 5.- Lines 107 and 117: mention to figures 2c and 2d seems to be messed up.
- 6.- Lines 117-123 and figures 2e and 2f: figure 2f is missing, which is probably the reason why, this part of the text is very difficult to follow and understand.
- 7.- Figure 2j: a histological micrograph of an untreated sample should also be added to enable comparison.
- 8.- Lines 180-181: this statement needs a reference.
- 9.- Figure 3g: which are the concentration units?
- 10.- Lines 244-245: the effectiveness of the thermal treatment depends not only on the temperature but also on the treatment time.
- 11.- Figure S5: being a_w one of the major parameters limiting food microbial growth, maybe this figure should not be supplementary, but present in the main manuscript.
- 12.- Line 321: authors state that they use 50 g of treated and untreated beef samples. If treated beef samples are of 1 cm³, this means roughly 1 g so, did authors collect up to 50 beef cubes to prepare each sample? In triplicate? And treated on the six sides of the cubes?

Reviewer #4 (Remarks to the Author):

This article presents a new food preservation process, applied to beef meat, consisting of carbonation of the meat surface through transient treatment at ultra-high temperature by Joule heating (until 2000 K). Microorganisms contaminating the surface are inactivated. The authors speak of molds, yeast and vegetative bacteria, but give no information on bacterial spores, such as *Bacillus cereus*, *Clostridium botulinum* or other pathogenic bacteria highly heat resistant. The very high temperature seems sufficient, but in a dry environment, the heat resistance of bacterial spores is greatly enhanced. As high temperature (over 373 K at atmospheric pressure) can only be achieved in dehydrated media, the resistance of bacterial spores needs to be assessed.

The authors show the microbial reduction on the surface (in a 0.1 mm layer). But what if the meat is contaminated in layer of more than 0.1 mm (if cut or pricked meat)? There is no guarantee the UFH treatment will keep food at room temperature without increasing the risk for the consumer.

Carbonation is associated with the production of neoformed compounds (NFCs), some of which are suspected carcinogens. Toxicological study is clearly insufficient to identify them. The discussion must warn against the lack of information given in this paper.

The rheological side is studied, but not the organoleptic (sensorial) side. However, the production of neoformed compounds can modified flavor and these NFC probably migrate into the meat. The authors have used questionable arguments to justify this method of food preservation in the first part of the introduction. It is true that the food insecurity is a major problem in the world (in one part of the world), while the food waste exists elsewhere. Could the shelf lives of foodstuffs (such as beef) have an impact on food insecurity in the world?

Furthermore, the carbonation of the surface of food leads to a loss of an significant part of the product if this thin layer is roughly removed by the consumer, which we can equate with food waste, contradicting previous arguments concerning food waste. Similarly, the reduction of energy use due to the refrigeration has to be set against with the waste due to the carbonated layer. It is not obvious the carbon footprint of refrigeration is greater than that of red meat. Please revise these argumentations.

Please revise the figure 2 (2f is missing).

Line 331-319 (Thermal transfer simulation): Water instantly turns to steam at 373 K (100°C) under atmospheric pressure. The vaporization temperature increases as water activity decreases, that's why the surface of the beef is dehydrated when it reaches 2000 K. Consequently, some of the water is vaporized during UFH processing. However, thermal simulations don't take this evaporation energy into account. Furthermore, simulations with 0.1 mm elements are inappropriate for simulating heat gradient around 1700 K in a layer of 0.1 mm.

Itemized list of response to reviewers' remarks

(Blue italic: Reviewers' remarks; Black type: Our response)

A. Reviewer #1

A.0. General remarks. *The article discusses the treatment of beef with ultrafast flash heating (UFH) to maintain the texture, quality, nutrition level, and visual appearance to increase the shelf life. The authors demonstrate that UFH can withstand bacterial spoiling for 5 days when stored at room temperature. The beef cube that was exposed to UFH undergoes some graphitization (~100 μm in thickness), but the rest of the meat is intact. The authors were able to control the heating time from 100 ms to 1400 ms to optimize the heating period that can yield the longest shelf life for the UFH-treated meat. The microbe analysis along with cytotoxicity states that the bacterial infection did not occur for 100 h. The interpretation of G' and G'' modulus indeed supports their claim that the meat quality was not changed after UFH treatment. Although the concept is very simple, it is effective and worth publishing in this journal. Here is a list of questions that must be considered. The reviewer is especially concerned on how they conducted the control experiment and how applicable this technique is to the food market industry and to a general household. Please see below.*

Our response: We appreciate the reviewer's positive comments. Specific questions raised by the reviewer are addressed as follows.

A.1. *Many chemical reactions (e.g., Maillard reaction) can occur during the heating process. Other than graphitization, what other chemical reactions occurred on the surface of the beef cube? Could the authors conduct more detailed chemical analysis to confirm the nature of surface after UFH? This is important to consider because we do not know whether the final product has some unwanted byproducts that might be detrimental to the consumer. For example, would it be possible to provide HPLC data of the surface of the beef cube after acid digestion?*

Our response: The reviewer has raised a critical question, and we agree that more thorough chemical analysis of the UFH-treated meat surface needs to be carried out. We would like first to clarify an important issue regarding the potential applications of the UFH method, which should help reconcile the concerns about the meat appearance and safety.

The UFH method is most useful for large-scale meat treatment, e.g., handling carcasses at slaughterhouses, since the cost-effectiveness is inversely proportional to the specific surface of the bulk meat. Large batch, long-distance meat transport requires cold chain storage, which would benefit from new freshness preservation technologies like the UFH treatment to lower the cost and help cope with unexpected temperature abuse. Then, the UFH-treated meat surface layer can be removed at the retailer's end and should not become a concern for customers.

However, we still agree that careful analysis of the UFH-treated meat is necessary for fundamental study purposes, and therefore we conducted high-performance liquid chromatography-mass spectrometry (HPLC-MS) and gas chromatography-mass spectrometry (GC-MS) of the material. Since the UFH-treated surface layer will not be consumed by customers, the analysis on acid-digested sample was not performed. We directly sampled the UFH-treated meat surface layer and carried out the HPLC and GC analysis.

For HPLC-MS, two different setups were used: an Orbitrap Q-Exactive mass spectrometer and a Waters Xevo TQ-XS triple quadrupole mass spectrometer (TQ-MS). The former was designed for high-resolution and accurate mass measurements suitable for the analysis of a wide range of analytes

in complex mixtures; and the latter is specialized for quantitative amino acid analysis. The results are summarized as follows.

A.1.1. Detection of acrylamide using TQ-MS. The TQ-MS results reveal the presence of acrylamide, at a concentration of < 7 parts per billion (ppb, $p < 0.05$). **Figure S3** has been added to the Supporting Information.

Figure S3. Total ion chromatogram (TIC) overview of acrylamide standard (a) and mass spectrum of acrylamide (b).

A.1.2. Chemical search using spectral libraries. The LC-MS results were processed using MZmine software, yielding 45 tentative matches. These matches were cross-referenced with several spectral libraries, including MassBank North America, MassBank Europe, the Human Metabolome Database, and NIST 14. This comprehensive approach enhances the reliability of the tentative identifications. The results are listed in **Table S1**, which is added to the Supporting Information.

Table S1. Spectral library search of the LC-MS results of the UFH-treated meat surface

Id	Area	~Retention time (RT)	Compound name	Matches cosine score
253	5.45E+07	0.3865	ACETYL-CARNITINE	0.999
587	5.27E+07	0.5829	ACETYL-CARNITINE	0.999
876	4.52E+06	1.2221	PROPIONYLCARNITINE	0.998
170	5.70E+07	0.375	CARNITINE	0.993
7394	6.15E+05	6.7092	Massbank:LU123902 N-Phenyl-1-naphthylamine N-phenyl-naphthalen-1-amine	0.992
7326	1.80E+06	6.6092	palmitoyl carnitine CollisionEnergy:205060	0.99
1156	5.89E+07	1.659	DL-PHENYLALANINE CollisionEnergy:102040	0.989
763	1.11E+07	0.7968	TYROSINE	0.989
407	4.86E+07	0.421	CARNOSINE	0.986

604	5.81E+06	0.5945	Methionine	0.985
956	3.18E+07	1.5061	INOSINE	0.984
626	2.62E+07	0.6293	HYPOXANTHINE	0.983
5315	4.29E+06	2.942	Massbank:RP025502 Hexanoyl-L-Carnitine[L-Hexanoylcarnitine](3R)-3-hexanoyloxy-4-(trimethylazaniumyl)butanoate	0.982
7611	5.87E+06	7.1046	PC(0:0/18:1); [M+H] ⁺ C26H53N1O7P1	0.982
7500	1.44E+07	6.9234	PC(0:0/16:0); [M+H] ⁺ C24H51N1O7P1	0.981
7342	9.71E+06	6.6208	PC(18:2/0:0); [M+H] ⁺ C26H51N1O7P1	0.98
7262	9.48E+06	6.4957	PC(18:2/0:0); [M+H] ⁺ C26H51N1O7P1	0.979
7680	1.17E+06	7.2036	PC(0:0/18:1); [M+H] ⁺ C26H53N1O7P1	0.978
7239	1.52E+06	6.472	PE(18:2/0:0); [M+H] ⁺ C23H45N1O7P1	0.978
706	3.07E+06	0.7125	xanthine CollisionEnergy:102040	0.975
7860	2.17E+06	7.6943	PC(0:0/18:0); [M+H] ⁺ C26H55N1O7P1	0.974
7482	1.25E+06	6.8995	PC(20:3/0:0); [M+H] ⁺ C28H53N1O7P1	0.972
7349	2.68E+06	6.6208	PC(0:0/20:4); [M+H] ⁺ C28H51N1O7P1	0.97
7568	3.73E+05	7.0395	PE(22:4/0:0); [M+H] ⁺ C27H49N1O7P1	0.97
5699	2.15E+05	3.1514	MassbankEU:SM817801 Cinnamamide(E)-3-phenylprop-2-enamide	0.969
7586	1.17E+06	7.0808	PE(18:1/0:0); [M+H] ⁺ C23H47N1O7P1	0.969
7533	5.12E+06	6.9588	PC(0:0/18:1); [M+H] ⁺ C26H53N1O7P1	0.968
949	6.87E+06	1.5061	HYPOXANTHINE	0.964
7314	1.90E+06	6.5975	PE(18:2/0:0); [M+H] ⁺ C23H45N1O7P1	0.96
802	3.10E+06	0.9104	3-hydroxybutyrylcarnitine	0.957
7176	1.66E+05	6.2373	PC(16:1/0:0); [M+H] ⁺ C24H49N1O7P1	0.95
7184	4.62E+05	6.2515	PC(20:5/0:0); [M+H] ⁺ C28H49N1O7P1	0.949
8478	2.05E+05	9.1848	cholesta-5,7-dien-3beta-ol	0.948
26	7.96E+05	0.3172	HISTIDINE	0.944
7656	2.86E+05	7.1762	PC(22:4/0:0); [M+H] ⁺ C30H55N1O7P1	0.942
6368	3.46E+05	3.9168	OCTANOYLCARNITINE	0.938
649	4.51E+06	0.6293	INOSINE 5'-MONOPHOSPHATE	0.926
6979	1.67E+05	5.3986	LAUROYLCARNITINE	0.906
8046	8.46E+05	8.1381	16-Hydroxyhexadecanoic acid CollisionEnergy:205060	0.897
7841	9.87E+05	7.6665	PE(18:0/0:0); [M+H] ⁺ C23H49N1O7P1	0.88
6759	1.12E+05	4.7271	DECANOYLCARNITINE	0.866
7472	7.26E+05	6.8726	PE(20:3/0:0); [M+H] ⁺ C25H47N1O7P1	0.838
853	1.25E+06	1.0939	Spectral Match to S-(5'-Adenosyl)-L-homocysteine from NIST14	0.824
846	1.45E+06	1.0812	Spectral Match to Glutathione, oxidized from NIST14	0.823

A.1.3. Tentative Matches from GC-MS Analysis. GC-MS was carried out to identify potential volatile and semi-volatile organic compounds, yielding 6 tentative matches. The results are listed in **Table S2**, which is added to the Supporting Information.

Table S2. Spectral library search of the GC-MS results of the UFH-treated meat surface

Compound match	~RT	Match
Glycerin	7.4	82.2P
4H-Pyran-4-one,2,3-dihydro-3,5-dihydroxy-6-methyl-	7.96	92.2P
Octanoic acid	8.03	72.5P
2(3H)-Furanone,dihydro-4-hydroxy-	8.14	95.9P
Decanoic acid	9.14	74.7P
Dodecanoic acid	10.19	75.3P

A.1.4. Heterocyclic aromatic amines (HAA) and benzo(a)pyrene were absent in the UFH-treated surface layer. No indication of the existence of HAA and benzo(a)pyrene was produced in the full scan analysis.

The following text has been added to the revised manuscript:

“Understandably, concerns about potential health risks due to the extremely high temperature treatment of the meat may arise. Therefore, we conducted a cytotoxicity assessment that indicated the UFH-treated surface does not show cell toxicity, since surface carbonization might be the dominant chemical process, as revealed by Raman spectroscopy (Fig. 21). Additionally, the generation of acrylamide, a carcinogenic compound to humans, is one of the major health concerns of cooking methods such as grilling and frying. High-performance liquid chromatography-mass spectrometry (HPLC-MS) analysis indicates that the acrylamide concentration remains at a level of < 7 parts per billion (ppb) in the UFH-treated beef surface (Fig. S3), which is far less than the 500-ppb benchmark limit set by the European Union regulation. Furthermore, an HPLC full scan analysis indicates that heterocyclic aromatic amines (HAA) and benzo(a)pyrene are absent from the sample (Table S1 and S2).” —L252-262

A.2. The comparison between the control and the UFH-treated meat is not a fair comparison. The authors directly used the meat that was stored at ~253 K, a normal freezer temperature. The chance that this meat was already in contact with bacteria is highly probable as there are no detailed experimental procedures that deal with the level of hygiene in preparing, cutting, processing, and shaping this meat into a cube. The control meat should also be pasteurized or treated once again to eliminate the germs before conducting the experiment.

Our response: Thanks for the careful check. We have provided details of the sample preparation: the key is that the samples used as control and for the UFH-treatment were prepared following the same procedure, to ensure a meaningful comparison. The following description was added to the Methods Section.

“Beef was purchased from a local retailer. To facilitate cutting and shaping, the beef was briefly stored in freezer (-20 °C) and then taken out and cut into cubes of 1 cm³. The beef cubes were then separated into two groups (used for the UFH treatment and control), each contained in a Ziplock bag, and were stored at -20 °C until the UFH experiments.” —L310-313

A.3. How is this technique more advantageous than a common cooking technique like sous vide or any non-destructive/non-chemical preservation methods (vacuum sealing)? For example, in sous vide, the meat is immersed in a water bath (about 50-70 degrees Celsius) to eliminate the bacteria and improve the texture. Of course, UFH technique might be faster, but it is impossible to incorporate this technique into a normal house since it requires a N₂-filled glove box and extremely high voltage. Or is this better for the factory/butcher shop, and not for homes?

Our response: The reviewer is correct: at this stage the UFH method is most useful for factories, not for the household, as we have clarified in response **A.1**. Additionally, this method should not be considered as a cooking technique, but rather a meat preservation and/or carcass decontamination technology. The following description has been added to the Discussion section.

“The ‘charred’ appearance of the UFH-treated meat may impact the acceptance by customers. Therefore, the UFH method may be most useful for large-scale meat treatment (e.g., handling carcasses at slaughterhouses) since the cost-effectiveness is inversely proportional to the specific surface of the bulk meat. After preservation and transport, the ‘charred’ surface layer (~100 μm in thickness) can be removed from the bulk meat at the retailer’s end to ensure the product is appealing to consumers. Future research may be conducted to avoid the carbonization effect by lowering the heating temperature and still maintaining the formation of the dehydrated surface layer.” —L263-270

A.4. The beef cube image shown in Figure 2h looks charred. Although the authors claimed that cutting the UFH-treated part, this might not be an attractive solution for the regular customer. For me, when I am buying a steak, I would be more inclined to buy the beef shown in Figure 2g.

Our response: The appearance of a food product indeed will significantly influence the customer’s will to purchase. Following the answers to **A.1** and **A.3**, we view UFH as an intermediate treatment carried out prior to retail distribution. The UFH method is more suitable for fulfilling the function of carcass preservation and/or decontamination.

A.5. The life cycle analysis of UFH treatment with a common pasteurization technique is required. Please include both refrigeration technique and vacuum sealing with subsequent heating to kill the germs.

Our response: Thank you for the observation and comment. The UFH technology is currently in a proof-of-concept stage, having been tested on small laboratory benchtop equipment which we have developed, as described in the manuscript. Therefore, the size and scope of the specific scaled-up application of the technology have yet to be determined. The technology could, conceptually, be used to treat small, primal or subprimal cuts of meat, or even entire animal carcasses, with commercial efficiencies, processing versatility, and consumer acceptance being among the main factors that would dictate its most appropriate and commercially feasible application. Due to these presently unknown factors, we feel it is not currently possible to do a life cycle analysis (LCA) that can be used to adequately and fairly compare the UFH technology to existing commercial technologies. We envision that as we continue to fine-tune the technology, these unknown factors will come into clearer focus, to the point that we will be able to perform a full LCA.

B. Reviewer #2

B.0. General remarks. *In general the manuscript is clearly written and appears to follow guidance on design and use of citations/references, as well as flow from the introduction to results, ending with Materials and Methods. The key findings are the microbiological and the color data of the treated beef, though the data are limited by lack of assessment directly of inoculated pathogenic microbes*

(e.g., Salmonella, E coli), thermo-tolerant/extreme-thermotolerant or otherwise, or the assessment of consumer liking/palatability of treated beef. The data should be significant, given its innovation and novelty. The authors do a good job at comparing to existing technologies, but the manuscript would benefit from similar comparisons for other foods (e.g., poultry, minimally processed produce, etc.). The methods appear to be sufficiently detailed to allow replication with sufficient research expertise. The reviewer would suggest the results be revised to shorten the first page's text to more quickly initiate the discussion of gathered data and its interpretations, versus the early text that does not move the reader into evaluation of findings.

Our response: Thanks for the positive feedback from the reviewer. This comment aligns with reviewer #4 (see response **D.4**). We have shortened the results discussion, particularly the description of Fig. 2 (see also our responses in **C.0** and **C.1**); also, we have the Introduction to communicate our message more quickly. The revised text is attached as follows.

“Meat is an important source of proteins, fatty acids, vitamins, as well as minerals such as iron and zinc, making it an important food source for human beings.¹⁻³ Global meat consumption has been growing steadily for several decades, exceeding 300 million tons since 2018.⁴ However, meat is also prone to spoilage due predominantly to the rich nutrient composition providing an excellent growth media for microorganisms, including food-born human pathogens and spoilage microorganisms. Thus, meat needs to be properly preserved and stored to ensure safe consumption, particularly during transport.

Meat preservation is typically achieved by storing the material at low temperatures (4 °C within 4 h of slaughtering), or by freezing (<-20 °C) or super-chilling in a partially frozen state. In general, low temperature methods have the advantage of maintaining better meat freshness compared to other techniques such as drying, smoking, and canning, or through chemical preservation (using salts, nitrites, acids, *etc.*) or biopreservation (using natural antimicrobial agents such as essential oils, nisin, and lysozyme, *etc.*).^{5,6} However, safely transporting meat requires energy to maintain the cold chain storage, which over long distance can be subject to disruption, risking spoilage and contamination with harmful microorganisms.

In this work, we demonstrate an ultra-high temperature flash heating (UFH) method that can efficiently preserve meats while retaining their nutrient level and texture. The concept stems from a similar ultra-fast and ultra-high temperature heating method that has been previously applied in materials science to sinter ceramics,⁷ synthesize high entropy catalysts,⁸ recycle plastics,⁹ and convert waste (including food waste),¹⁰ *etc.* For the first time, we have adapted this novel technology for food preservation. In this approach, Joule heating is achieved by applying a high electric power (~1,800 W) over a short time interval (< 1 s) to a carbon substrate, creating a transient increase of the substrate temperature to > 2,000 K. By placing meat in direct contact with the carbon substrate, this rapid heating causes ultra-fast dehydration and inactivation of microorganisms on the meat surface, forming a thin (~100 μm) protective layer that also serves as a barrier to inhibit bacterial migration into the material. As a result of the effects of surface dehydration and microorganism-inactivation after the application of UFH, we achieved an extended shelf life of 5 days at room temperature for beef storage without sacrificing the interior appearance, texture, and nutrition of the meat. The UFH method is essentially a food surface treatment technology, which could be cost-effective for industrial-scale preservation of large carcasses. This UFH approach could be used alone or in combination with low temperature technologies to decrease the risk of temperature change during transport and storage for improved food safety.”—L38-70

C. Reviewer #3

C.0. General remarks. *The manuscript shows an interesting and novel approach to preserve meat (and possibly other food products) by means of flash heating. There are some major concerns related to the methodology and to the research itself, which should be properly addressed before the manuscript reaches publication.*

The main issue is related to the methodology itself. Beef in particular and food in general does not show regular surfaces. Plenty of protuberances, holes, crevices or even porous surfaces can be found in any piece of food, which will not easily adapt to the carbon felt. If, as authors claim, the flash heating method only reaches 0.1 mm depth, then microorganisms present in this sort of hiding places could find protection and escape the treatment. If the heated surface reaches deeper, then carbonization may be too extensive and may harm too much food quality. In one way or another, the proposed technology may be of little use for the food industry. There are some other major concerns and minor remarks that authors should address. Major concerns:

Our response: The reviewer has raised a general concern that is of pivotal importance. We would first like to address the efficacy of the UFH method when applied to irregular (porous, rough, etc.) meat surfaces, and then address point by point the specific questions that follow.

A rough surface can also be effectively heated using our UFH method, as the heat transfer mechanism is governed by *radiation transfer* rather than direct contact transfer. The reviewer's concern may arise from confusion due to our original thermal analysis, in which we made an approximation that the heating element and the meat surface are 'thermally bonded'; and as a result, fine details of the temperature change of the meat surface during heating did not come into our calculation. We have conducted a completely new thermal analysis in this revision, considering radiation is the main heat transfer mechanism that causes the temperature of the beef surface to increase. The results are shown in the updated Fig. 2 (Fig. 2d and 2e). It can be seen from Fig. 2d that the meat surface temperature can reach ~900 K during the UFH treatment, with the heating element temperature profile shown in Fig. 2c, which is sufficient for sterilization. Due to the radiation heat transfer mechanism, tight contact between the heating unit and the meat surface is not critical, so pathogens 'hiding in' surface pores can still be inactivated.

Note that since the contact between the heating element and the meat surface is not ideal, the temperature of the meat surface is significantly lower than that of the heating element, but is still able to reach ~900 K. If needed, the meat surface temperature can be increased by increasing the heating element temperature (we can achieve temperatures of >3,000 K using the Joule heating concept of the UFH process, as we have demonstrated for different purposes, e.g., nanoparticle synthesis). But such temperatures may be unnecessary, as several of the reviewers have already pointed out, the temperature used in the current manuscript may already have been too high. Our future work may focus more on exploring the low temperature limit that would allow us to achieve a similar sterilization effect without impacting the meat surface quality as much.

The description of Fig. 2d in the main text has been revised, quoted as follows. The caption has also been updated.

"We conducted a two-dimensional numerical simulation to investigate the transient temperature distribution of the beef sample during the UFH process. To simplify the simulation, we assumed that only one face of the beef was in direct contact with the carbon felt heating element, with the other face being exposed to an air environment at 293 K. The initial bulk temperature was set at 253 K, which is typical for long-term food storage. The simulation results indicate the beef surface temperature

increased from ~450 K at 0.1 s to the peak point of ~900 K in 1 s during UFH, then quickly dropped down below ~400 K in 5 s, after the heating was stopped (Fig. 2d). Our simulation also showed that at distances of > ~2 mm from the heated surface, the beef temperature was almost unchanged.” — L103-111

Fig. 2. Analysis of the UFH treatment of meat. (d) Thermal analysis results of the evolution of the beef temperature over 10 s of UFH treatment. The simulation sensors are along the central line normal to the carbon felt, from the heated beef surface across the entire cube.

The following text has been added to the Discussion section. The thermal analysis methodology in the Methods has also been updated accordingly.

“It needs to be pointed out that thermal radiation is the dominating heat transfer mechanism in the UFH method, as evidenced from the high light emission of the heating element (Fig. 2b). As a result, the UFH treatment does not require perfect contact between the meat surface and the heating element, allowing it to sterilize the material even when the meat surface is rough. Additionally, the UFH method can be scaled up since the heating unit is flexible and can effectively cover expansive areas with irregular surfaces, such as large carcasses, with imperfect contact being overcome by the radiative heat transfer.” —L232-238

C.1. *Figure 2d shows the temperature profile of the carbon felt during the treatment. It starts measuring the temperature at about 1500 K and stops measuring, after the treatment, at about 1300 K. Both of them are extremely high temperatures. What does it happen before and mainly after the measured interval? Probably the increase in temperature is very fast, but usually cooling is not so fast and, from the plot, a “tail-kind” behaviour could be predicted, i.e., temperature could remain very high for a long time. And this is for the carbon felt. What about the piece of meat?*

Our response: Thanks for this important question. The temperature profile of the *carbon felt* was recorded by a high-speed thermal camera, based on the brightness of the images (a thermocouple could also provide temperature measurements, but does not feature a rapid enough response). As a result, 1500 K is approximately the highest temperature that can be accurately measured using this method. It is clear that the temperature can reach ~1500 K in ~0.2 s.

The ‘tail effect’ of the meat surface after the heating is stopped needs to be addressed. As shown in our new thermal analysis (**Fig. 2d** in response in **C.0**), in 5 s the meat surface temperature has dropped to a temperature of < ~400 K, and in 10 s, the heating effect ceases entirely. The temperature of the interior of the meat (at distances of > ~2 mm from the heated surface) remained changed.

The effective heated layer of ~0.1 mm represents the condition when the meat surface was UFH treated for ~ 1 s and was fully cooled down. Note that once the heating is ceased, the surface temperature change of the meat surface follows a different mechanism of convection heat transfer, which explains the rapid temperature drop after heating. In this regard, the UFH meat surface treatment is fundamentally different from conventional cooking, in which the entire piece of meat is cooked. The following text has been added to the Discussion section.

“As seen in Fig. 2d, temperature variation only takes place on the meat exterior (~2 mm from the surface); and the interior temperature remains unchanged throughout the process. When heating is ceased, the inward heat transfer from the meat surface relies solely on thermal conductance, and the heat is quickly dissipated to the environment via convection (Fig. S10). In this regard, the UFH meat surface treatment is fundamentally different from conventional cooking, in which the entire piece of meat is cooked.” —L245-251

C.2. *Figures 3d and e: authors show growth curves reaching 10 log CFU/g or even more but have performed only 5 dilutions (line 323) and have countable ranges up to 250 CFU/plate (line 326). This means that the maximum log count they can reach is about 3×10^7 CFU/g.*

Our response: Thanks for the careful check. For the long-term microbiological test, we used dilutions of 10^{-3} , 10^{-6} , 10^{-9} , 10^{-12} and 10^{-15} . The Methods section has been revised accordingly:

“The UFH treated and control samples, 5 g each, were collected, immersed in 450 mL of sterile buffered peptone water (BPW), and stomached for 2 min using a stomacher (Seward type 80, UK). Additional four dilutions of 10^{-2} , 10^{-3} , 10^{-4} , and 10^{-5} were successively prepared. For the long-term test, five additional dilutions of 10^{-3} , 10^{-6} , 10^{-9} , 10^{-12} and 10^{-15} were prepared.” —L326-329

C.3. *Carbonization: authors claim that carbonization might impact the acceptance of consumers. Actually, the piece of meat shown in figure 2h looks like quite unpleasant to me. Authors argue that, to avoid such carbonization effect, the heating temperature could be lowered. Then, why did not they try with a lower temperature in this very research?*

Our response: Thanks for raising this important question. The appearance of the UFH-treated meat is a general concern. This method is more suitable for large-scale treatment prior to retail (see our response to **A.1**, **A.3**, and **A.4**). The reviewer also suggests performing similar experiments at lower temperatures. We agree that it will be a meaningful study to systematically examine the temperature effect to determine the boundary at which this preservation/decontamination method is still effective, while also avoiding unwanted chemical byproducts, such as acrylamide, as well as significant color change. In this contribution, however, our focus was to prove the concept that flash heating may effectively form a thin sterilized protective layer, and may eliminate (or reduce the amount of) harmful

chemicals such as acrylamide, heterocyclic aromatic amines (HAA), and benzo(a)pyrene, as detailed in response to **A.1**.

C.4. *Cytotoxicity: authors test cytotoxic effects of their treatment but it would be probably more advisable to measure the formation of Maillard reaction compounds, such as acrylamide.*

Our response: Thanks for the suggestion. We conducted comprehensive HPLC-MS and GC-MS analysis on the UFH-treated meat surface. Please see response **A.1** for details.

C.5. Minor remarks.

C.5.1. *The objective of the research is not clearly stated. Although it can be easily drawn from the context, it should be properly written.*

Our response: We have rewritten the Introduction to clarify the research objective. Please see response **B.0**.

C.5.2. *Figure 1a: it does not show the apparatus and processing, but a piece of meat.*

Our response: Fig. 1a is a conceptual illustration of the meat experiencing the UFH treatment: the bright substrate represents the heating unit (carbon felt). The following text has been added to the caption of Fig. 1 to clarify the description.

“The thickness of the outer layer with respect to the bulk sample is not to scale.” —**L92-93**

C.5.3. *Figure 1e: if the cube is 1 cm³ volume, then the picture roughly shows a surface layer of almost 0.5 mm width, not 0.1 as claimed by the authors.*

Our response: Thanks for the careful check. The entirety of Fig. 1 is a conceptual illustration. The thickness of the UFH-treated layer was not to scale to facilitate visualization. Please see response **C.5.2**.

C.5.4. *Lines 91-92: how is the treatment applied to the six faces of the cube? Is it a discontinuous process that has to be applied in three different times?*

Our response: As a proof-of-the-concept experiment, the heating of the 6-faces of a meat cube was conducted separately. However, the process can be made continuous due to the flexibility of the carbon felt. In response **C.0**, we have shown that radiation heating is the dominant heat transfer mechanism of the UFH process, which makes non-contact continuous heating possible. Systematic study of this processing method would be the subject of future work.

C.5.5. *Lines 107 and 117: mention to figures 2c and 2d seems to be messed up.*

Our response: Thanks for the careful check. We have corrected the error. Also, Fig. 2d and 2e have been updated, representing the results of our new heat transfer analysis.

C.5.6. *Lines 117-123 and figures 2e and 2f: figure 2f is missing, which is probably the reason why, this part of the text is very difficult to follow and understand.*

Our response: Thanks for the careful check. We have re-numbered the sub-figures in Fig. 2 and made the according changes to the main text.

C.5.7. *Figure 2j: a histological micrograph of an untreated sample should also be added to enable comparison.*

Our response: Please see supporting materials, Fig. S7.

C.5.8. *Lines 180-181: this statement needs a reference.*

Our response: Reference [24] has been added. Thanks for pointing this out.

C.5.9. *Figure 3g: which are the concentration units?*

Our response: The unit is $\mu\text{g/mL}$, we have updated Fig. 3g, shown below:

Fig. 3g. Cell viability of 3T3-L1 (mouse embryonic fibroblast cell) and CCD18co (human colon tissue cell) at different concentrations of meat sampled from the surface of the UFH-treated beef.

C.5.10. *Lines 244-245: the effectiveness of the thermal treatment depends not only on the temperature but also on the treatment time.*

Our response: Thanks for noting this subtle point. Our original description, "...is significantly higher than required to eliminate all types of bacteria, including *E. coli* and *Salmonella*, where a temperature of $>343\text{ K}$ ($70\text{ }^\circ\text{C}$) is considered effective...", might cause confusion. The effectiveness of our UFH method is based on the results of our microbiology tests, not on the direction comparison of temperatures with other methods. We deleted the sentence for clarity purposes.

C.5.11. *Figure S5: being aw one of the major parameters limiting food microbial growth, maybe this figure should not be supplementary, but present in the main manuscript.*

Our response: Thanks for the suggestion. We obtained more detailed results on the water content change during and after the UFH treatment, and included it in Fig. 2, shown below. Fig. S5 remains as supporting data.

Fig. 2. Analysis of the UFH treatment of meat. (e) Thermal analysis results of the evolution of the beef water content (mass fraction) over 10 s; the simulation sensors are along the central line normal to the carbon felt, from the heated beef surface across the entire cube.

C.5.12. *Line 321: authors state that they use 50 g of treated and untreated beef samples. If treated beef samples are of 1 cm³, this means roughly 1 g so, did authors collect up to 50 beef cubes to prepare each sample? In triplicate? And treated on the six sides of the cubes?*

Our response: We appreciate the careful check. It is actually 5 g. The Methods section has been updated.

D. Reviewer #4

D.1. *This article presents a new food preservation process, applied to beef meat, consisting of carbonation of the meat surface through transient treatment at ultra-high temperature by Joule heating (until 2000 K). Microorganisms contaminating the surface are inactivated. The authors speak of molds, yeast and vegetative bacteria, but give no information on bacterial spores, such as Bacillus cereus, Clostridium botulinum or other pathogenic bacteria highly heat resistant. The very high temperature seems sufficient, but in a dry environment, the heat resistance of bacterial spores is greatly enhanced. As high temperature (over 373 K at atmospheric pressure) can only be achieved in dehydrated media, the resistance of bacterial spores needs to be assessed.*

Our response: Thanks for your valuable feedback. We acknowledge the importance of considering spore-forming bacteria in the context of food safety. In response to your comment, we would like to clarify our position regarding the scope of our current research. The experimental setup and focus of our study were primarily designed to explore the effects of transient ultra-high temperature treatment on non-spore-forming microorganisms. Considering the technical complexities and different

mechanisms of spore resistance, more thorough investigation on spore inactivation would require a substantially different experimental design. We hope to explore this aspect in follow-up studies.

D.2. *The authors show the microbial reduction on the surface (in a 0.1 mm layer). But what if the meat is contaminated in layer of more than 0.1 mm (if cut or pricked meat)? There is no guarantee the UFH treatment will keep food at room temperature without increasing the risk for the consumer.*

Our response: The reviewer raises an important point, which relates to applicability of the UFH method. We have clarified (A.1 and A.3) that the UFH method would work best when applied at the level of slaughterhouse, to treat bulky carcasses prior to transport and subsequent packaging. It would not be efficient for treating cut, pricked, or ground meat.

D.3. *Carbonation is associated with the production of neoformed compounds (NFCs), some of which are suspected carcinogens. Toxicological study is clearly insufficient to identify them. The discussion must warn against the lack of information given in this paper. The rheological side is studied, but not the organoleptic (sensorial) side. However, the production of neoformed compounds can modified flavor and these NFC probably migrate into the meat.*

Our response: Thanks for the suggestions. We have added HPLC-MS and GC-MS data. Please see response A.1 for details.

D.4. *The authors have used questionable arguments to justify this method of food preservation in the first part of the introduction. It is true that the food insecurity is a major problem in the world (in one part of the world), while the food waste exists elsewhere. Could the shelf lives of foodstuffs (such as beef) have an impact on food insecurity in the world? Furthermore, the carbonation of the surface of food leads to a loss of an significant part of the product if this thin layer is roughly removed by the consumer, which we can equate with food waste, contradicting previous arguments concerning food waste. Similarly, the reduction of energy use due to the refrigeration has to be set against with the waste due to the carbonated layer. It is not obvious the carbon footprint of refrigeration is greater than that of red meat. Please revise these argumentations.*

Our response: We agree that the first part of the introduction has little relevance to the main theme of this paper. These sections have been rewritten in the Introduction to better refine our research motivation. Please see response B.0 for details.

D.6. *Please revise the figure 2 (2f is missing).*

Our response: Thanks for the careful check. Fig. 2 has been updated.

D.7. *Line 331-319 (Thermal transfer simulation): Water instantly turns to steam at 373 K (100°C) under atmospheric pressure. The vaporization temperature increases as water activity decreases, that's why the surface of the beef is dehydrated when it reaches 2000 K. Consequently, some of the water is vaporized during UFH processing. However, thermal simulations don't take this evaporation energy into account. Furthermore, simulations with 0.1 mm elements are inappropriate for simulating heat gradient around 1700 K in a layer of 0.1 mm.*

Our response: The reviewer is correct: a more refined thermal transfer simulation should consider the phase change of water. We have conducted this new calculation, in which details of the heat transfer mechanisms (radiation, contact, and convection), as well as the temperature change due to phase change enthalpy were considered. Simulation details have been included in the Supplementary Information, quoted as follows.

Supplementary Method. Analysis of the temperature and water content change of the beef sample during the UFH treatment.

Figure S10. Schematic diagram of heat and water vapor transport during ultra-high temperature exposure of frozen beef at $-20\text{ }^{\circ}\text{C}$.

At the initial time ($t = 0$), a carbon heater, powered by a direct current (DC) supply and located at the bottom of the frozen beef, generates a high temperature of 2000 K lasting for 1 second. This heat is transferred to the frozen beef, potentially increasing its temperature, melting the ice, and causing the water within the beef to evaporate. Concurrently, the beef exchanges heat with its surroundings mainly through convection and radiation. By assuming uniformity in the beef, the variation of the temperature ($T(t, x, y)$) and water content ($C(t, x, y)$) within it can be approximately described as follows:

$$\rho C_p \frac{\partial T(t, x, y)}{\partial t} = k \nabla^2 T(t, x, y) \quad (1)$$

$$\frac{\partial C(t, x, y)}{\partial t} = D \nabla^2 C(t, x, y) \quad (2)$$

where ρ (kg/m^3) is the density of the beef, C_p ($\text{J}/(\text{kg}\cdot\text{K})$) denotes the heat capacity of the beef, k indicates the temperature-dependent thermal conductivity^[1], and D represents the diffusion coefficient of the water^[2]. Here we assume that water mass transfer in the beef does not initiate until the beef is unfrozen, meaning ice sublimation and mass transfer can be neglected before the beef melts ($D = 0\text{ m}^2/\text{s}$). We employed the apparent heat capacity method to solve the heat transfer equation (Eq. (1)), assuming that beef melting occurs within a specific temperature interval. The latent heat of melting is taken into account by the modification of the specific heat capacity, which is expressed by:

$$c_p = \theta_s c_{p,s} + \theta_l c_{p,l} + L_{s \rightarrow l} \frac{\partial \alpha_m}{\partial T} \quad (3)$$

where θ_s and $c_{p,s}$ represent the volume fraction and specific heat capacity of the beef before melting, respectively; θ_l , and $c_{p,l}$ represent the volume fraction and specific heat capacity of the beef before melting. $L_{s \rightarrow l}$ is the latent heat of melting, and α_m is defined as the mass fraction of the melted beef. Assuming that water evaporation occurs only on the surface of the beef, the boundary conditions of the heating surface and other surfaces for heat transfer equation (Eq. (1)), including convective, radiative, and evaporative heat transfer, can be described as follows, respectively:

$$k\nabla T = h(T_{air} - T) + \varepsilon\sigma(T_h^4 - T^4) + DL_{l \rightarrow g}\nabla C \quad (4)$$

where h ($W/(m^2 \cdot K)$) is the convection heat transfer coefficient, T_{air} (K) and T_h (K) denote the surrounding air temperature and the heater temperature, respectively, $L_{l \rightarrow g}$ is defined as the molar latent heat of vaporization, ε refers to the surface emissivity of the beef, σ is the Stefan-Boltzmann constant, and $DL_{l \rightarrow g}\nabla C$ denotes the heat flux out due to moisture vaporization. According to the conservation of mass, the boundary condition for mass transfer equation (Eq. (2)) is as follows:

$$D\nabla C = k_c(C_b - C) \quad (5)$$

where C_b denotes the air moisture concentration and $k_c = h_m/(\rho_{p,l} \cdot C_m)$ is defined as the mass transfer coefficient of the water. In this context, h_m is mass transfer coefficient in mass units and C_m is the specific moisture capacity. In the simulations, the initial temperature (T_0) is set at 253.15 K, and the initial water concentrations of the beef is roughly estimated by $C_0 = \varphi\rho_{p,l}/M_{H_2O}$, where $\varphi = 0.75$ represents the water content. The surrounding air temperature is 293.15 K and the convection heat transfer coefficient is 13.75 $W/(m^2 \cdot K)$. More details can be found in Table 1.

Table S1. List of the parameters used for simulation.

Parameters	Value	Description
T_{air}	293.15 K	surrounding air temperature
T_0	253.15 K	initial temperature of the beef
h	13.75 $W/(m^2 \cdot K)$	convection heat transfer coefficient
M_{H_2O}	0.018 kg/mol	water molecular weight
C_0	43542 mol/m ³	initial moisture concentration
C_b	1161.1 mol/m ³	air moisture concentration
C_m	0.003	specific moisture capacity
h_m	1.67×10^{-6} kg/(m ² ·s)	mass transfer coefficient in mass units
k_c	5.33×10^{-7} m/s	mass transfer coefficient
$L_{l \rightarrow g}$	41400 J/mol	molar latent heat of evaporation
$L_{s \rightarrow l}$	180.8 kJ/kg	latent heat of melting
$\rho_{p,l}$	1045 kg/m ³	density of melted beef
$\rho_{p,s}$	961 kg/m ³	density of frozen beef
$c_{p,l}$	3510 J/(kg·K)	specific heat capacity of melted beef
$c_{p,s}$	2090 J/(kg·K)	specific heat capacity of frozen beef

Figure S11. Simulated (a-c) temperature distribution and (d-f) relative water content distribution in the heated beef.

References

- [1] Willix J, Lovatt S J, Amos N D. Additional thermal conductivity values of foods measured by a guarded hot plate[J]. *Journal of Food Engineering*, 1998, 37(2): 159-174.
- [2] Trujillo F J, Wiangkaew C, Pham Q T. Drying modeling and water diffusivity in beef meat[J]. *Journal of Food Engineering*, 2007, 78(1): 74-85.

“

REVIEWERS' COMMENTS

Reviewer #1 (Remarks to the Author):

The chemical composition of meat after the ultrafast flash heating (UFH) treatment demonstrates that no highly toxic chemicals were detected according to high-pressure liquid chromatography-mass spectrometry (HPLC-MS) and gas chromatography-mass spectrometry (GC-MS) analysis. This crucial data indicates that the UFH treatment of meat could be useful in increasing the longevity of the meat quality without imposing health threats. Especially, the absence of heterocyclic aromatic amines (HAA) and benzo(a)pyrene, two widely common carcinogens formed during heating process of meat, in both surface and internal part of meat also supports the authors' claim that UFH is an efficient process for preserving meat. The detailed control experiment procedure in the Methods section of the manuscript cleared the concern of how the experiment was conducted. The reviewer agrees that the UFH treatment is currently in the proof-of-concept stage, and the but a preliminary life cycle assessment (LCA) seems like it is required now by many editors to assess the viability of any new proposed effort. But I leave that decision to the editor and I wish not to contest this point further, please. For future work, the reviewer recommends including LCA that includes two important cost evaluation steps: the cost of UFH treatment per square foot of the animal carcass and labor fees to remove UFH-treated surface of the meat. The reviewer recommends the manuscript for publication in Nature Communications.

Reviewer #2 (Remarks to the Author):

The reviewer is appreciative of the authors' efforts to clarify various points raised by the reviewers, and supply additional data, explanation, or correction to the manuscript. The reviewer senses the manuscript is improved in its revised form, and adds additionally useful data explaining toxicology and mechanistics of flash heating. The reviewer does question the likelihood of whole carcass treatment, despite the authors' claims that irregular surfaces are adaptable. The design, application, and adaptability to animal carcasses of differing sizes makes that difficult to envision.

However, the manuscript is better in its updated form, and the reviewer believes the authors have satisfied needed changes.

Reviewer #3 (Remarks to the Author):

Authors have made an important effort to properly answer the queries of the reviewers, even performing additional research. However, some important concerns still persist.

C0: Authors claim now that the method is intended to treat big meat pieces such as carcasses. Carcasses contain muscle packs, bones, etc., showing even more uneven surfaces, with not just pores or holes, but also wide piece separations (much wider than 2 mm) which will be more difficult to reach with this technology, even when the temperature is not changed at or below 2 mm from the surface (line 246).

C2: Why performing so many dilutions? Actually, the most important question is: how did authors get so high bacterial loads, as high as 10^{12} CFU/g

C.5.2 minor remark has not been properly addressed

C.5.12: then the dilution -1 is not -1 but almost -2.

Additional concerns: if UFH treatment is considered as a preservation treatment, then meet should not be previously frozen, since this would affect the quality of the meet, and UHF is supposed to preserve meet efficiently without further preservation treatments.

In the supplementary simulation of the temperature and water content change of the beef sample during UFH treatment authors assume that the temperature of the surrounding air is room temperature, but beef will be surrounded by the carbon felt, which will be at about 2000K, and will not let the air, water, etc. escape, so the surrounding temperature will be closer to this 2000 K. Please, revise.

Reviewer #4 (Remarks to the Author):

The authors have sufficiently answered my questions, even if it would have been preferable to mention certain remarks in the final manuscript, for example, the fact that the impact of the UFH process was not studied on bacterial spores.

Itemized list of response to reviewers' remarks (*Blue italic: Reviewers' remarks*; Black type: Our response)

A. Reviewer #1

The chemical composition of meat after the ultrafast flash heating (UFH) treatment demonstrates that no highly toxic chemicals were detected according to high-pressure liquid chromatography-mass spectrometry (HPLC-MS) and gas chromatography-mass spectrometry (GC-MS) analysis. This crucial data indicates that the UFH treatment of meat could be useful in increasing the longevity of the meat quality without imposing health threats. Especially, the absence of heterocyclic aromatic amines (HAA) and benzo(a)pyrene, two widely common carcinogens formed during heating process of meat, in both surface and internal part of meat also supports the authors' claim that UFH is an efficient process for preserving meat. The detailed control experiment procedure in the Methods section of the manuscript cleared the concern of how the experiment was conducted. The reviewer agrees that the UFH treatment is currently in the proof-of-concept stage, and the but a preliminary life cycle assessment (LCA) seems like it is required now by many editors to assess the viability of any new proposed effort. But I leave that decision to the editor and I wish not to contest this point further, please. For future work, the reviewer recommends including LCA that includes two important cost evaluation steps: the cost of UFH treatment per square foot of the animal carcass and labor fees to remove UFH-treated surface of the meat. The reviewer recommends the manuscript for publication in Nature Communications.

Our response: Thanks for the reviewer's supportive comments. We agree that CLA is an important component of research associated with sustainability; and would like to pursue it when our UFH technology becomes more matured in the future publications. It is, however, worth of making a remark on this; the following sentences are added in the last paragraph.

“A full life cycle analysis, however, is not available, because it depends heavily on multiple factors, such as the application targets (though in principle this method is most energy efficient for large meat pieces such as carcasses, smaller products such as primal or subprimal cuts of meat are other options) and meat distribution scheme, etc. The heating device need to be optimized to satisfy various application needs.”—Line 281-286.

B. Reviewer #2

The reviewer is appreciative of the authors' efforts to clarify various points raised by the reviewers, and supply additional data, explanation, or correction to the manuscript. The reviewer senses the manuscript is improved in its revised form, and adds additionally useful data explaining toxicology and mechanisms of flash heating. The reviewer does question the likelihood of whole carcass treatment, despite the authors' claims that irregular surfaces are adaptable. The design, application, and adaptability to animal carcasses of differing sizes makes that difficult to envision. However, the manuscript is better in its updated form, and the reviewer believes the authors have satisfied needed changes.

Our response: Thanks for supporting our work in the revised manuscript. We appreciate the reviewer's understanding of the difficulties related to applying the UFH method to large meat pieces. This work is still in the proof-of-the-concept stage; and much need to be done toward scaling up. In this case, as has been correctly pointed out by the reviewers, the major challenge lies in the *macroscopic* unevenness of the meat surface. We've demonstrated the *microscopic* working mechanisms of the UFH method. Filling the gap of applying a technology at multiple scales is always

one of the biggest engineering challenges. It is foreseeable, however, that a viable solution would be to combine multiple technologies. For example, memory material might be combined with the heating element in the UFH method in the future, serving as a shaping scaffold, and taking advantage of the flexible nature of the carbon substrate, to achieve a close contact with the meat. As long as the distance between the heating element and the meat surface is reduced to millimeter scale (ideally in close contact), owing to the radiation heating mechanism as we described in the manuscript, effective heating can be accomplished. Such method of course will impose complex engineering problem on its own, and is beyond the scope of the current paper.

C. Reviewer #3

C.0. General remarks. *Authors have made an important effort to properly answer the queries of the reviewers, even performing additional research. However, some important concerns still persist.*

Our response: Thanks for accepting our efforts in previous response. We make further elucidation in the following.

C.1. *Authors claim now that the method is intended to treat big meat pieces such as carcasses. Carcasses contain muscle packs, bones, etc., showing even more uneven surfaces, with not just pores or holes, but also wide piece separations (much wider than 2 mm) which will be more difficult to reach with this technology, even when the temperature is not changed at or below 2 mm from the surface (line 246).*

Our response: Thanks for the comment, which is consistent with the one raised by reviewer #2. We addressed the scaling-up concern in **Section B**. Here we would like to clarify the statement of the heat transfer distance, namely, after heating the interior meat (>~2mm from the surface) temperature remains unchanged.

Under the condition that meat surface has been effectively heated, our thermal analysis suggests that the interior meat temperature is almost unchanged. This is due to the fact that thermal conductance of meat is very low, and the heating on the surface is only applied in a short period. Heating the meat *surface*, however, relies on radiation heat transfer; and the low conductance of meat does not play a role. In this regard, close contact between meat surface and heating unit is not a prerequisite (but of course, the closer, the better). As a result, to apply the UFH method to an irregular surface, one should focus on how to let the carbon substrate be adaptive to the macroscopic surface. In **Section B**, we suggest that such effect may be achieved by combining multiple technologies, including employing a memory material support, which invites future investigations.

C.2. C2. *Why performing so many dilutions? Actually, the most important question is: how did authors get so high bacterial loads, as high as 10^{12} CFU/g?*

[Figures 3d and e: authors show growth curves reaching 10 log CFU/g or even more but have performed only 5 dilutions (line 323) and have countable ranges up to 250 CFU/plate (line 326). This means that the maximum log count they can reach is about 3×10^7 CFU/g.]

Our response: Thank you for the detailed check. We included a series of dilutions in our microbial testing protocol to ensure that the bacterial concentrations fell within the quantifiable range of our counting methods. The unusually high bacterial loads of 10^{12} CFU/g observed in our samples may be attributed to the factors specific to our study context. The main reason is that our control group sample was stored in room temperature for ten days, during which high levels of bacterial contamination were expected.

C.3. *C.5.2 minor remark has not been properly addressed*

[Figure 1a: it does not show the apparatus and processing, but a piece of meat.]

Our response: Thanks for the comment. To improve Fig. 1a we add annotations, and the caption has been updated, see below. The UFH process is highly dynamic, and is difficult to show in figure. We add a video clip to the Supplementary Information (Supplementary Movie S1), recorded using a high-speed camera, in a collection rate of 200 fps, showing the entire heating process. Hope these efforts resolve the concern of the reviewer.

Fig. 1a. ...Demonstration of the heating process. Carbon felt is used as the heating element, due to its high electrical resistance. The arrow marks the electric current direction.

C.4. *Additional concerns: if UFH treatment is considered as a preservation treatment, then meet should not be previously frozen, since this would affect the quality of the meet, and UHF is supposed to preserve meet efficiently without further preservation treatments.*

Our response: Thanks for point this out. Our ultimate goal is to use the UFH method to enhance the safety and quality of meat products, regardless of their initial state. In our paper, using frozen meat as a model system, this method immediately suggests an application in preventing temperature abuse of meat products during transport/storage. Thus, it is an *auxiliary* preservation method in the current context. To clarify, we add the following sentence to the main text:

“Note that at this stage UFH serves as an auxiliary meat preservation method, since frozen meat was used as a model; however, no significant obstacles are foreseen to its application to fresh meats.”—
line 255-257

C.5. *In the supplementary simulation of the temperature and water content change of the beef sample during UFH treatment authors assume that the temperature of the surrounding air is room temperature, but beef will be surrounded by the carbon felt, which will be at about 2000K, and will not let the air, water, etc. escape, so the surrounding temperature will be closer to this 2000 K. Please, revise.*

Our response: Thanks for the careful check. Our current simulation setting is based on the UFH experiment performed, as shown in Fig. 1a and the Supplementary Media. The main purpose of the thermal analysis is to show how the UFH affect the interior meat temperature. In this regard, whether or not the non-heated surface is exposed to the environment plays but a minor role, since the heat

transfer toward the interior is limited, due to the poor thermal conductance of meat; and more importantly, to the flashing heating itself—the heating source is ceased in ~ 1 s.

D. Reviewer #4

The authors have sufficiently answered my questions, even if it would have been preferable to mention certain remarks in the final manuscript, for example, the fact that the impact of the UFH process was not studied on bacterial spores.

Our response: We thank the reviewer for the supportive comments. To be rigorous, we add the following sentence to our main text. More microbiological tests will be carried out in the future studies.

“Current microbiological tests do not involve bacterial spores.”—175-176.